# Civil war and death in Yemen: Analysis of SMART survey and ACLED data, 2012–2019

Debarati Guha Sapir[1,2☉], Jideofor Thomas Ogbu[1☉], Sarah Elizabeth Scales[3‡*], Maria Moitinho de Almeida[1‡], Anne-Francoise Donneau[4‡], Anh Diep[4‡], Robyn Bernstein[5‡], Akram al-Masnai[6‡], Jose Manuel Rodriguez-Llanes[7‡], Gilbert Burnham[2‡]

1 Centre for Research on the Epidemiology of Disasters, Institute of Health and Society, Université catholique de Louvain, Brussels, Belgium, 2 Department of International Health, Johns Hopkins University Bloomberg School of Public Health, Baltimore, Maryland, United States of America, 3 Epidemiology Program, College of Health Sciences, University of Delaware, Newark, Delaware, United States of America, 4 Public Health Department, Biostatistics Unit, University of Liege, Liege, Belgium, 5 CDC Zambia, Lusaka, Zambia, 6 Oxfam GB, Oxford, United Kingdom, 7 European Commission, Joint Research Centre, Ispra, Italy

☉ These authors contributed equally to this work.
‡ These authors also contributed equally to this work.
* sescales@udel.edu

**Data Availability Statement:** CE-DAT data is available from the Centre for Research on the

## Abstract

Conflict in Yemen has displaced millions and destroyed health infrastructure, resulting in the world's largest humanitarian disaster. The objective of this paper is to examine mortality in Yemen to determine whether it has increased significantly since the conflict began in 2015 compared to the preceding period. We analysed 91 household surveys using the Standardized Monitoring and Assessment of Relief and Transitions methodology, covering 2,864 clusters undertaken from 2012–2019, and deaths from Armed Conflict Location & Event Data Project database covering the conflict period 2015–2019. We used a Poisson-Gamma model to estimate pre-conflict ($\mu_p$, baseline value) and conflict period ($\mu_c$) mean death rates using household survey data from 2012–2019. To analyse changes in the distribution of deaths and estimate nationwide excess deaths, we applied pre- and post-conflict death rates to total population numbers. Further, we tested for association between excess death and security levels by governorate. The national estimated crude death rate/10,000 in the conflict period was 0.20 (95% CI: 0.17, 0.24), which is meaningfully higher than the estimated baseline rate of 0.19 (95% CI: 0.17, 0.22). Applying the conflict period rate to the Yemeni population, we estimated 168,212 excess deaths that occurred between 2015 and 2019. There was an 17.8% increase in overall deaths above the baseline during the conflict period. A large share (67.2%) of the excess deaths were due to combat-related violence. At the governorate level, posterior crude death rate varied across the country, ranging from 0.03 to 0.63 per 10,000 per day. Hajjah, Ibb, and Al Jawf governorates presented the highest total excess deaths. Insecurity level was not statistically associated with excess deaths. The health situation in Yemen was poor before the crisis in 2015. During the conflict, intentional violence from air and ground strikes were responsible for more deaths than indirect or non-violent causes. The provision of humanitarian aid by foreign agencies may have helped contain increases in indirect deaths from the conflict.

Epidemiology of Disasters (CRED) upon reasonable request. Armed Conflict Location and Event Data Project (ACLED) data is available for download at https://acleddata.com/data-export-tool/. A datafile is provided with the supplementary materials.

**Funding:** The authors received no specific funding for this work.

**Competing interests:** The authors declare no competing interests.

## Introduction

The 2019 United Nations Office for Coordination of Humanitarian Affairs' (UNOCHA) Humanitarian Needs Overview on Yemen describes the plight of a country that, despite the severity of the humanitarian crisis, is infrequently at the top of news headlines or global agendas. However, more than 24 million Yemenis, or 80% of the population, require some type of humanitarian assistance, and 19.7 million lack adequate health care, including emergency services, routine vaccination, and maternity care [1]. The war has pushed nearly 10 million people to the brink of starvation, 2 million of whom are children with acute malnourishment. Further, 222 out of the 333 districts in the country are already one step away from famine [2,3]. Within two years of the declaration of the conflict, 50% of children under five years of age were stunted due to severe and sudden food insecurity [4]. By the end of 2017, health facility deaths and recorded injuries directly attributable to conflict–just the tip of the iceberg for these metrics–continued to mount [5].

News coverage of this devastation was largely ignored until Yemen was embroiled in the largest cholera outbreak in modern history. Major newspapers, (e.g., Washington Post, BBC, CNN) began reporting the outbreak at the end of August 2017, although suspected cases had already started to accelerate in April, 5 months earlier. The cumulative total from October 2016 to November 2019 reached over 2 million suspected cholera cases and 3,886 associated deaths, a case fatality rate of 0.17% [6]. Only beginning in May 2018, more than 18 months after the beginning of the outbreak, did the first cholera vaccine campaign start in Yemen. However, this devastation is not new. In 2016, the Global Burden of Disease Study (GBD) reported "conflict and terror" as the second cause of death and leading cause of premature death in Yemen [7].

The Yemeni crisis features a complex network of both internal and external forces, with multiple layers of conflict–with distinct ideological, geopolitical, sectarian, and separatist interests–occurring simultaneously [8–10]. The current Yemeni conflict became internationalized when Saudi Arabia and the Gulf states, with arms support from the United States and United Kingdom, began aiding the largely Sunni forces [11]. Concurrently, Iran has increasingly supported the Zaydi Houthi forces. Since 2015, Yemen has been subjected to nearly 20,000 air/drone strikes–an average of 4,000 per year. This bombardment has severely damaged the already fragile and dysfunctional health facilities and destroyed supply chains for food and medicine [12]. These impacts are further exacerbated by the ongoing blockade of foreign ports–particularly Sanaa–that have severely constrained food and aid imports into the country. Economically, Yemen–the poorest country in the Eastern Mediterranean World Health Organization (WHO) region–has largely collapsed. Foreign exchange reserves have been depleted and foreign remittances diminished [13]. The Government is now unable to pay for imports of food and other commodities and cannot meet public sector salaries.

Utilization of health care and preventive services such as immunisation or maternal services have declined, and child health indicators (e.g., on-schedule immunisation, anthropometric status, anaemia, diarrhoeal disease incidence) have worsened since 2016 [14–16]. El Becherar-oui et al., observed that some governorates with high levels of violence displayed worse health indicators [17]. Despite over five years of war, there have been few attempts to estimate excess mortality, and these rare efforts were made difficult by dysfunctional health information reporting or civil registration and vital statistics (CRVS) systems [18].

However, accurate mortality rates are central to inform policy and action priorities starting from, humanitarian response, credible advocacy, and short-term resource programming. It is equally important for social justice and reconciliation once peace is established. Further, estimations of human losses are crucial to legal prosecutions and reparation cases such as the DR

Congo versus Uganda case in front of the International Court of Justice [19]. In this paper, we examine overall mortality in Yemen, exploring whether it has increased significantly in the period of the current conflict (2015–2019) compared to the immediate pre-conflict period. In the absence of credible vital statistics, we use Bayesian statistical methods to analyse small-scale surveys undertaken by humanitarian organisations. In addition, we examine the relationship between posterior crude mortality rates and conflict insecurity at governorate level using analysis of variance test to ascertain how mortality to ascertain how mortality at governorate level varies across the different insecurity level.

## Materials and methods

### Data sources

Collecting mortality data is a major challenge in conflict-affected countries. Death certification is rare. Vital events registration systems are one of the first victims of a failed state and death tolls remain obscure [20,21]. The current data published by the United Nations Statistics Division Demographic Statistics, shows that in 2007 just 12% of deaths had been recorded in Yemen's death registry [21]. To circumvent the lack of official mortality data, we aggregated mortality statistics from 2,864 individual geographical clusters in 91 small scale mortality surveys collected by six organisations.

For this study, we utilized surveys from the Complex Emergency Database (CE-DAT), a repository of small-scale surveys undertaken by humanitarian agencies and the United Nations Office for Humanitarian Aid (UN-OCHA) [20]. The UN data include surveys undertaken by the World Food Programme (WFP), the Food and Agriculture Organisation of the United Nations (FAO), the Yemeni Ministry of Health (MoH), and the surveys done for the UN Emergency Food Security and Nutrition Assessment process (see S2 Table for characteristics of surveys included in the study). These surveys were chosen for analysis because they provide the most recent mortality data at the highest resolution. All surveys used a standard two-stage random cluster sampling method, used the Standardized Monitoring and Assessment of Relief and Transitions (SMART) methodology, and provided at least one estimate per governorate from 2012 to 2019 [22]. SMART surveys primarily collect information on nutrition and mortality (all cause and age-specific) indictors as well as information on vaccination coverage, food insecurity, and access to water and sanitation in a targeted population. It is important to note that more than half of the surveys used reported excluding areas due to inaccessibility or insecurity. For governorates without population values, we calculated denominators using

**Table 1. Summary characteristics of data sources for the period 2012–2019, Yemen.**

| Variable | Source | Indicator | Description |
|---|---|---|---|
| Number of deaths | CE-DAT & UNOCHA | Mortality | Total no. of deaths within the recall period of the survey. We used the number as reported in each survey or estimated based on mortality information presented in the reports. |
| Sample size | CE-DAT & UNOCHA | Mortality | Mortality sample size used in calculating crude death rate. We used the number as reported or estimated based on available information in the report. |
| Recall period | CE-DAT & UNOCHA | Mortality | Mortality recall period as used in the survey, in days or months. |
| Population estimate | MoH, UNICEF, World Bank, WorldPop & WorldOmeter | Population | Governorate population from 2015–2019 projected based on 2012–2014 average pop. growth values in MoH and UNICEF reports. Country level population for 2015–2019 from World Bank and WorldOmeter. Population values were adjusted using IOM displacement data for each governorate. |
| **Deaths from war violence** | ACLED & Yemen Data project | Violence | No. of deaths attributable to violent events related to conflict such as battles, explosions, remote violence, protests |
| Air/drone strike | ACLED | Violence | No. of air/drone strikes reported by ACLED |

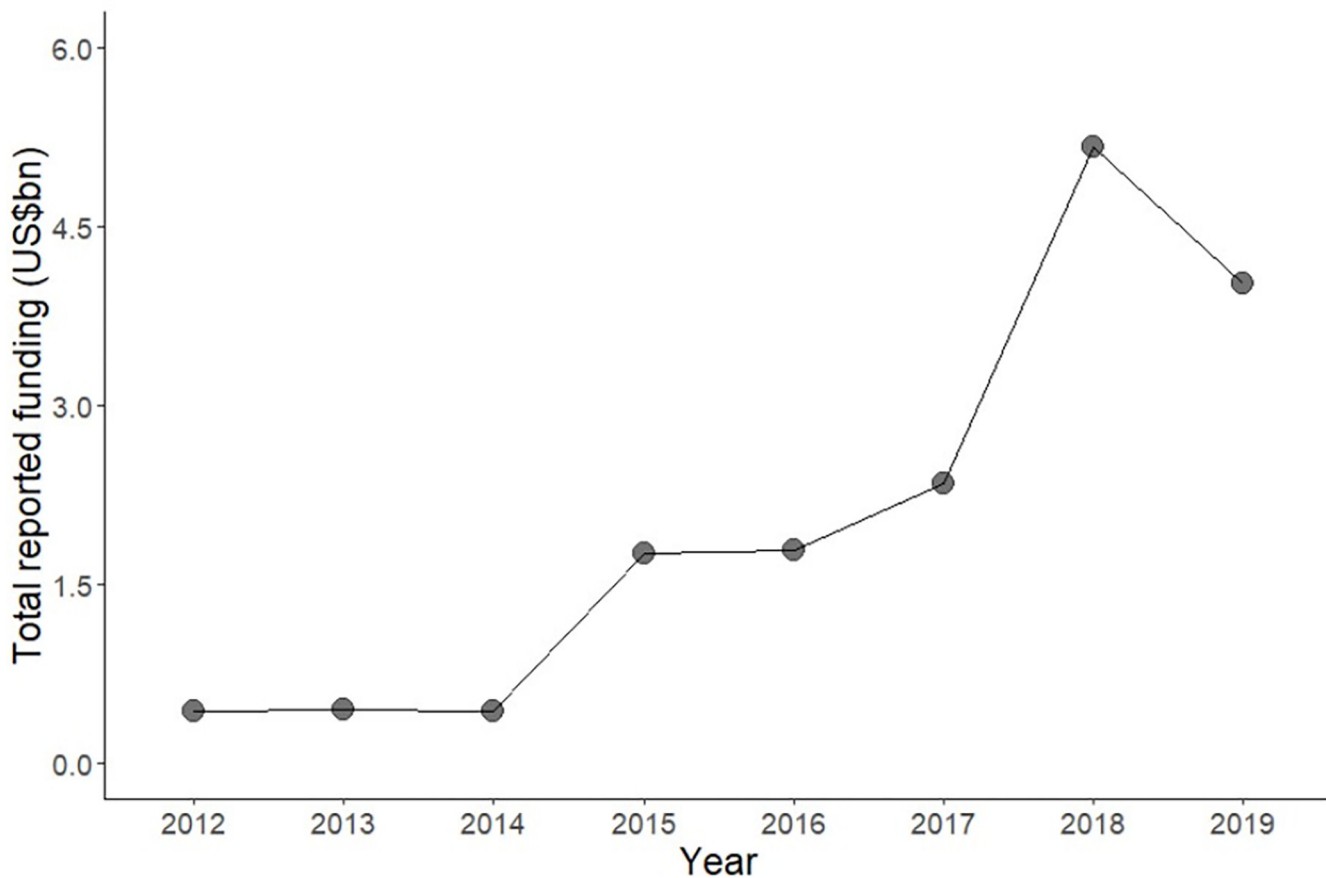

**Fig 1. Survey selection flowchart, describing the survey selection process, inclusion/exclusion criteria, and number of surveys at each selection stage.**

population growth data from 2015 to 2019 and adjusted for displacement with data from UNICEF and the International Organization for Migration (IOM) (Table 1) [23].

The earliest time point for which there were surveys which provided all the necessary data and used the SMART methodology was 2012. In total, we retained 91 surveys (28 from 2012–2014, and 63 from 2015–2019). The most common mortality information presented in the survey reports are for all-cause mortality are number of all-cause deaths, crude death rate (CDR), number of people included in sample, number of sampled households. Information on number of people that joined/left the household within the recall period is also provided in the report. See S2 Table for information on surveys. The Crude Death Rate is the number of people in a total population who died within a specified period of time. The unit of measure in SMART survey is deaths/10000/day. We extracted information on the number of deaths, crude death rate, mortality recall period and sample sizes from the surveys. In the absence of number of deaths, we approximated the estimate as follows: number of deaths = (CDR*recall period*sample sizes)/10 000. We combined the extracted data from the surveys to obtain our analysis dataset.

The selection process is described in Fig 1.

## Mortality model

We used a standard approach to estimate excess mortality thus we established the national baseline crude death rate (BCDR) using surveys conducted in the three years preceding the

2015 crisis declaration, and then the posterior crude death rate (PCDR) from surveys conducted between 2015–2019. The difference is defined as the "total excess mortality." The BCDR is the pre-conflict posterior mean estimates ($\mu_p$, baseline value) and the conflict period PCDR ($\mu_c$) estimates is the conflict period posterior mean estimate.

For the mortality estimate we extracted information on the number of all cause deaths, the recall period, and the sample size/targeted population. The outcome variable which is the number of all caused death within a recall period is a count variable and we assumed it follows a Poisson distribution. To estimate the pre-conflict and conflict period estimates we divided our dataset into two main periods: pre-conflict 2012–2014 and conflict period 2015–2019. We fitted a Bayesian Poisson-Gamma model, which allows us to capture the complex structure of the data by considering possible over-dispersion in the data [24]. We included person-days as an offset variable to account for difference in sample sizes and recall periods. Person-day calculation requires information on births and changes in household composition within a specified recall period of that survey (e.g., 3 or 12 months). However, since complete demographic profiles were not available for surveys, we approximated person-day values using the estimated population and mortality recall period [22].

To obtain the BCDR of the CDR ($\mu_p$) and the PCDR of the CDR ($\mu_c$), we used a hierarchical mixture of the Poisson-gamma model (Eq (1)) for modelling the crude number of deaths obtained in the surveys ($i = 1,2,3\ldots, m$), where $m = 28$ (number of surveys in pre-conflict period) or 63 (number of surveys in the crisis period). We modelled the crude number of deaths in each survey using a Poisson distribution with parameter $\lambda$:

$$n_i \sim pois\ (\lambda_i)$$

$$\lambda_i = pt_i * \theta \tag{1}$$

$$\theta \sim Gamma(\alpha, \beta)$$

where, $n_i$ is the number of observed deaths in survey i, $pt_i$ is the person-days for survey $i$. The choice of priors for the pooled mean μ and the gamma parameter—β are a non-informative gamma prior of the gamma form $(0 \cdot 001, 0 \cdot 001)$, where α is derived as the product of the pooled mean μ and β. We ran 100,000 iterations with a burn-in of 50,000 to sample from the distribution via the Markov Chain Monte Carlo simulation. We checked convergence and assessed stationarity using Geweke diagnostics [25].

We estimated total excess deaths (TED) for the whole country for the period 2015–2019 using the formula below:

$$TED = \left\{ \left(\mu_c - \mu_p\right) * \left(\frac{365}{10000}\right) * n * median.adj.tot.pop \right\} + tot.direct\ deaths \tag{2}$$

$\mu_c$ = posterior mean CDR for conflict period /10000/day—(PCDR)
$\mu_p$ = posterior mean CDR for pre-conflict period /10000/day—(BCDR)
$n$ = number of years (2015–2019)
$median\ tot.pop$ = median of the adjusted total population
$tot.\ direct.deaths$ = total direct ACLED deaths from 2015–2019
$median\ tot.pop$ = median of the adjusted total population
$tot.\ direct\ deaths$ = total direct ACLED deaths from 2015–2019
$(\mu_c) * \left(\frac{365}{10000}\right) * n * median.adj.tot.pop$ is the observed number of deaths
$(\mu_p) * \left(\frac{365}{10000}\right) * n * median.adj.tot.pop$ is the expected number of deaths
$Total\ cumulative\ deaths = (\mu_c) * \left(\frac{365}{10000}\right) * n * median.adj.tot.pop + tot.direct\ deaths$

For governorate-level estimates, we estimated the PCDR. For each governorate with a single reported CDR between 2015–2019, we used this as the reported PCDR. These were calculated using the formula below:

$$\sum_{j=1}^{22} \left\{ \left[ (\alpha c_j - \mu_p) * \left( \frac{365}{10000} \right) * (n * \text{median.adj.pop}_j) \right] + DD_j \right\} \tag{3}$$

$$\alpha c_j \begin{cases} \mu_c \text{ posterior CDR (PCDR) at governorate level} \\ \omega c_j \text{ reported conflict period CDR when only one value is available} \end{cases}$$

median.adj.pop$_j$ = median population by governorate
DD$_j$ = direct violent deaths by governorate
The model building process is summarised in S1 Fig.

## Insecurity classification index method

Since 2015, in addition to ground warfare with gunshots and explosives, Yemen absorbed a minimum of 18,481 air/drone strikes, although not all strikes were reported [12]. The most targeted governorates were Sa'dah (4,932), Sana'a (2,013), Taizz (1,978), Hajjah (1,945), and Al Hudaydah (1,755) (S1 Table).

Globally, human security indices are commonly generated by combining multi-dimensional indicators such as jailed populations, the number of homicides, ease of access to small arms, and political instability rankings [26,27]. In Yemen, such data are, in most part, missing. We therefore used publicly available data on violent deaths attributable to air/drone strikes and armed events (e.g., battles, explosions/remote violence, protests, and riots) to create a proxy index that broadly reflects the level of insecurity using the following formula:

$$\text{Security Score} = \sum_{q=1}^{n} \left[ \frac{e_{pqr} * F_{pqr}}{E_{qr}} \right] \tag{4}$$

$$\text{Insecurity Index} = \inf\{x \in R : \ g \leq F(\text{Security Score})\} \tag{5}$$

The insecurity index is classified according to quantiles: 0.25 quantile as Insecure; 0.5 quantile as Serious; 0.75 quantile as Severe; and 1 quantile as Extreme. Where, *epqr* is the number of events q in governorate p for year r. *Eqr* is the sum total of events q in all governorates in year r, *Fpqr* is the total fatality from events q in governorate p for year r, and the Security Score is the sum of weighted fatalities in governorate p for year r, where n = 6 [28]. We matched the insecurity index obtained for each governorate for each year with the corresponding PCDR to obtain the analysis data set. To understand how the different levels of conflict insecurity in Yemen affect the overall mortality, we assessed associations between the PCDR in governorates and their insecurity index using a one-way ANOVA test. We used R Studio version 4.0.2 and packages R2jags version 0.6.1, stats, HD Intervals version 0.2.2 for our analyses. All p values were two-tailed with one degree of freedom and $\alpha = 0\cdot05$.

## Results

The death rate in the conflict period (PCDR) was slightly higher than the baseline rates (BCDR) and had tighter confidence interval. The overall nation-wide excess death rate was 0.03/10,000 population. Often when baselines are not available in conflict settings, regional average mortality rates proposed by SPHERE Guidelines are used [29]. In this case, the 2011

**Table 2. Excess death rate estimates using baselines and conflict period CE-DAT surveys, Yemen, 2015–2019.**

|  | Rate estimates (10000/day) [LCL, UCL] [a] |
|---|---|
| **Total Crude Death Rate (2015–2019)** | 0.23 [0.19, 0.27] |
| **Posterior Crude Death Rate (PCDR)** | 0.20 [0.17, 0.24] |
| Baseline Crude Death Rate (BCDR) | 0.19 [0.17, 0.22] |
| Direct (Violent) Death rate | 0.02[b] |
| Total Excess Death Rate (TED) | 0.03 [0.00, 0.07] |

[a] LCL: Lower credible level; UCL: Upper credible level.

[b] No confidence interval provided. Data source: www.acleddata.com.

SPHERE baseline was the average death rate in the North Africa and Middle East region i.e., 0.16—a rate that is relatively low as most of the countries are in middle or upper-middle income economic categories. In this study, we used the survey-based baseline (BCDR) as those that provided the best available estimates that reflected the specific realities of Yemen. The excess death rate–the difference between PCDR and BCDR–is presented below (Table 2).

Applying the conflict period rates (PCDR) to the Yemen population, we estimate 1,115,024 deaths would have occurred in the 5-year crisis period, whereas we would have expected 946,812 deaths had the estimated baseline rate (BCDR) prevailed—a 17.8% increase in deaths. Excess deaths attributable to direct and indirect causes linked to the crisis led to 168,212 more deaths, or an average of 92 excess deaths per day in the population. Of these, 67.2% were attributable to direct combat related violence and the remainder were attributable to indirect causes. At the governorate level, PCDR varied across the country ranging from 0.11 to 0.35 per 10,000 per day. The excess deaths for each governorate are calculated from their excess death rates based on their population. Hajjah, Ibb, Aden, Saada, and Al Jawf present the highest total excess deaths. Hadramaut and Al Maharah, both sparsely populated, had some of the lowest rates along with Al Hudaydah, Dhamar and Lahj (Fig 2).

The insecurity index was highest for the three governorates (Hajjah, Al Hudayda, Taizz) along the coastline (Fig 3) and those controlled by the Houthi faction. Sana'a was especially targeted, experiencing 4,932 attacks—about 1,000 attacks per year, on average. Sa'adah, Al Bayda, Aden and Al Dali were also exposed to high levels of violence and air strikes. All governorates in the mainland experienced air/drone strikes (S1 Table).

We found no evidence for differences in death rates between governorates that were extremely insecure compared to those with lower levels of insecurity (Fig 4).

## Discussion

Since the beginning of the civil war in 2015, humanitarian organizations and the press have reported high death tolls in Yemen [30–33] but reported qualitatively as "thousands" or "tens of thousands" of deaths. Undoubtedly many have died, but little statistical evidence has been presented to support increased death rates, reflecting the challenges the country faces in mortality surveillance [33].

When calculating a credible excess mortality estimate, the choice of baseline is an important challenge. For example, two well-known studies from Iraq used different baselines, leading to almost 100,000 deaths (5.5/1,000) in one and 405,000 (2.9/1,000 person-years) in the other [34,35]. A similar situation emerged from using different baselines when estimating mortality in the Second Congolese War (1998–2004) [36]. Excess mortality in Darfur was calculated based on World Bank reports (0.3/10,000/day), producing widely differing results from estimates by Reeves [29,37–40]. We chose to construct a baseline from locally implemented

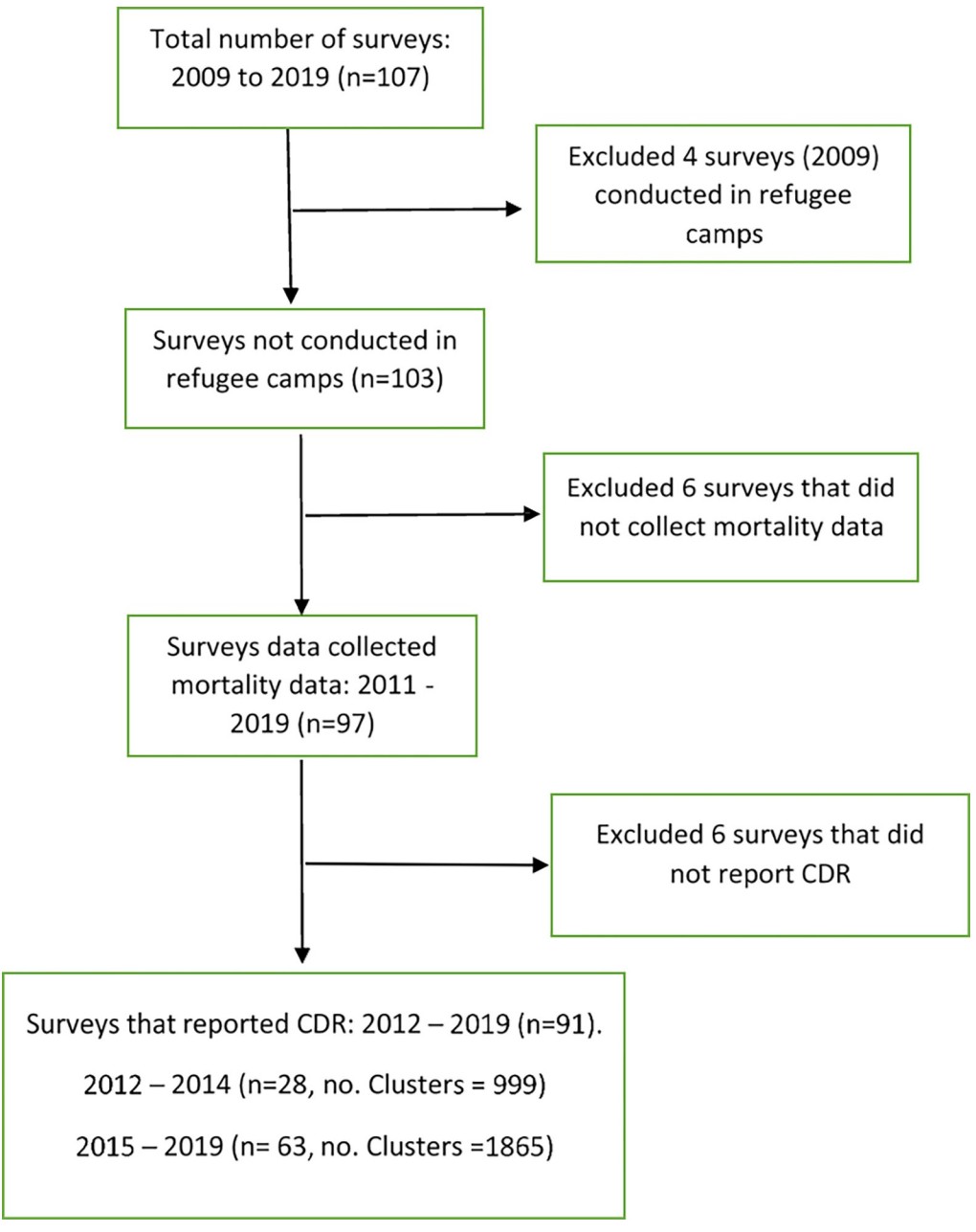

**Fig 2. Geographical distribution of total excess deaths based on excess death rates by governorate, Yemen, 2015–2019.** Source: Shape files extracted from Global Administrative Areas (2012). GADM database of Global Administrative Areas, version 2.0. [online] URL: www.gadm.org. Created using QGIS version 3.10.3.

mortality surveys from the pre-crises period, which provides a timely and more precise representation of mortality in Yemen.

Our results suggest that death tolls since 2015 are greater than would be expected, based on the estimated pre-war value. However, the conflict period death rate was not strikingly higher than the levels experienced by the country in the years preceding the current crisis. Prior to the beginning of the 2015 conflict, the Yemeni health system had already started a downward slide where health system broke down progressively and health status deteriorated unlike more concentrated violence and massacre witnessed in DRC [41–43].

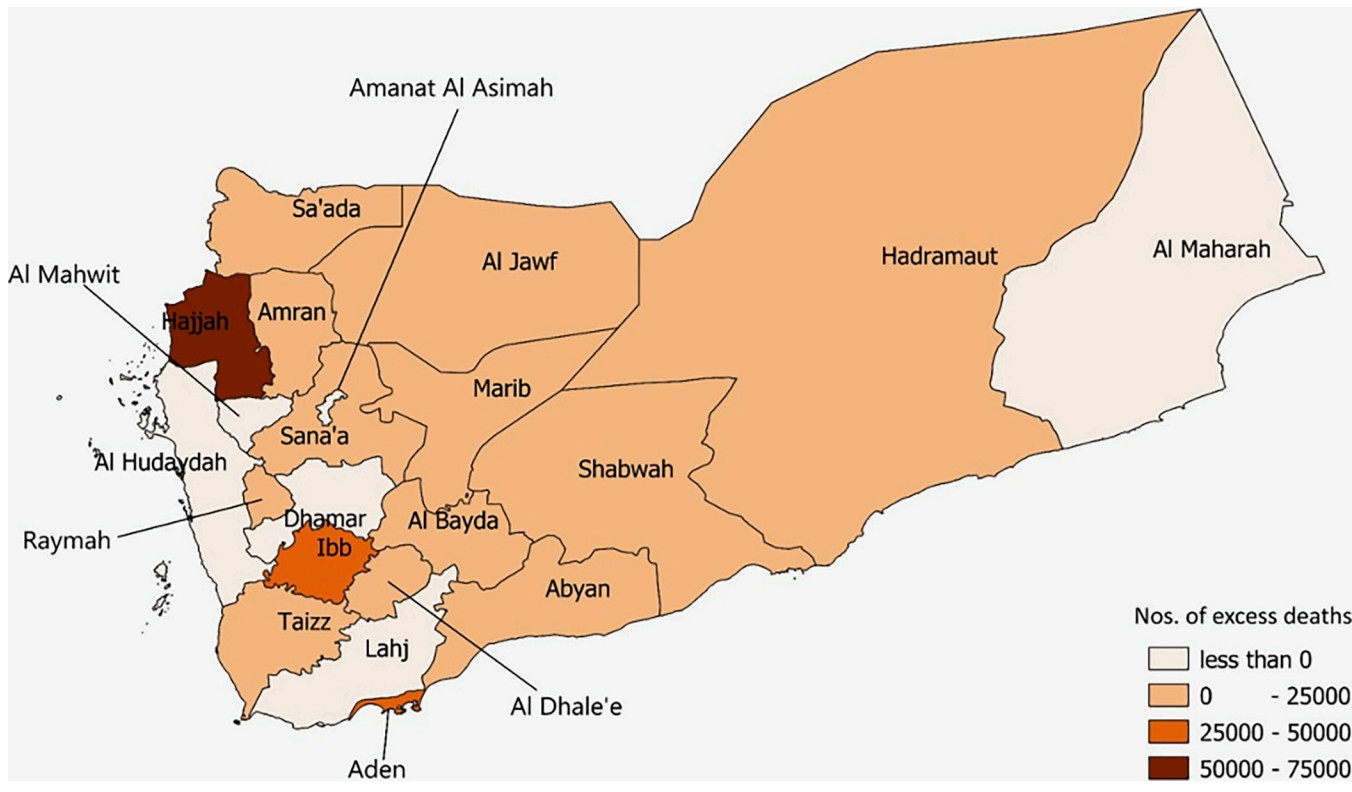

**Fig 3. Geographic distribution of insecurity index levels, Yemen, 2015–2019.**

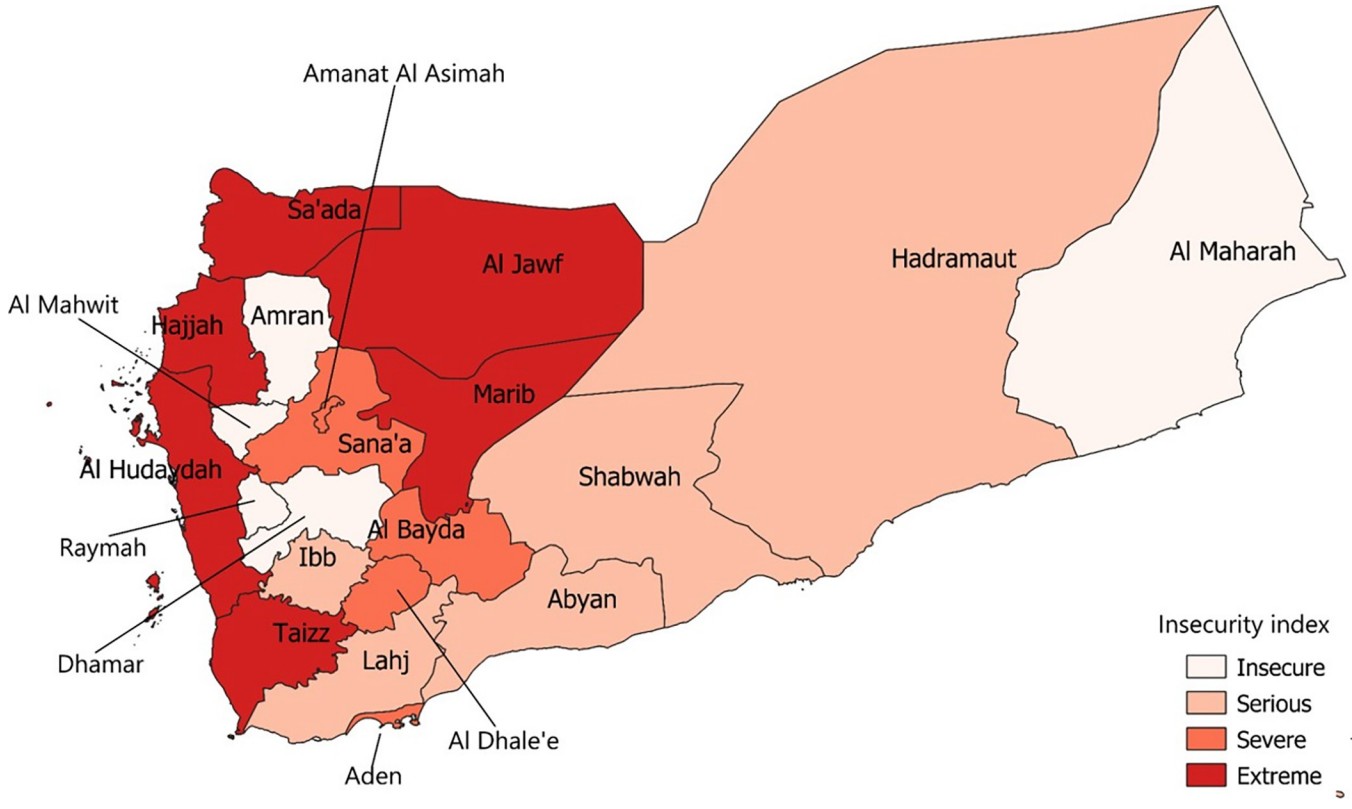

**Fig 4. Posterior crude death rate by governorate and insecurity level, Yemen, 2015–2019.**

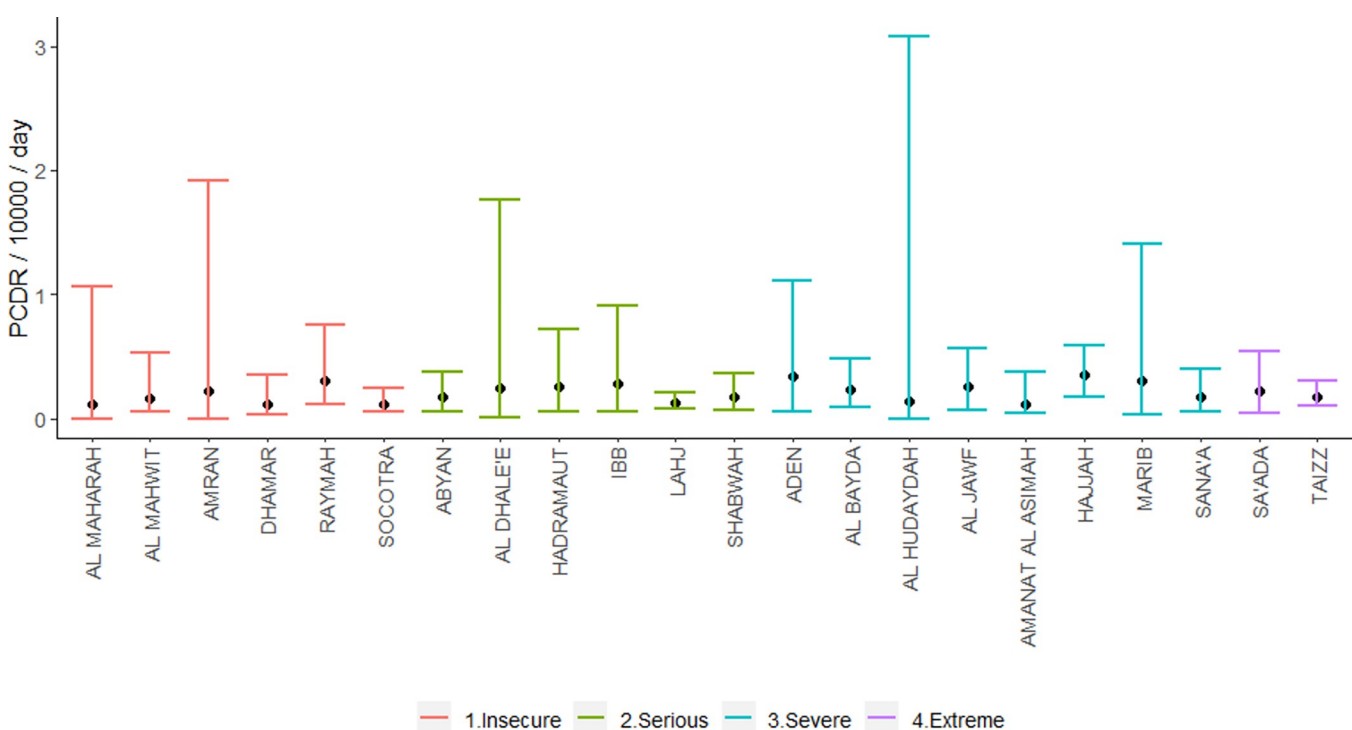

**Fig 5. Humanitarian aid assistance to Yemen 2012–2019.** Data source: United Nations Office for the Coordination of Humanitarian Aid (UN-OCHA) [52].

Second, a large share of excess deaths was attributable to armed violence, as recorded in ACLED and the YDP–another noticeable departure from scenarios seen in other civil conflicts where indirect deaths were by far the largest proportion of deaths [44–46]. A possible explanation is the protracted nature of the conflict in Yemen. Our data and analysis seem to portray a conflict where indirect excess mortality is persistently higher at constant levels, relative to regional averages. Therefore, a sudden spike in violence by the use of different weaponry, mostly air bombardments, produces a larger contribution of direct violent deaths total excess mortality.

Third, our index of human insecurity was not correlated with the estimated conflict period deaths, which were similar in all governorates regardless of their level of insecurity. The mere threat of air strikes and violence may have been enough to halt essential health care and other critical life-sustaining activities across the country, not just in governorates specifically targeted with intense air strikes and fighting.

We found the total number of deaths that occurred in Yemen in the five years of crisis was 17.8% higher than expected–nearly 170,000 additional deaths, or about 34,000 per year of conflict on average. Other conflicts, such as in Iraq, Darfur, Syria, and the Democratic Republic of Congo have reported higher conflict-related deaths [34,36,37,47]. Despite appeals from the humanitarian community, no intervention was mounted, and overseas aid remained low (Fig 5). As early as 2004, the census reported a crude death rate of 7.5/1,000 population, substantially higher than the UN regional average of 5.3/ 1,000 population [48]. A decade later, data from the Demographic and Health Survey (DHS) reported child mortality of 53/1,000 live births, nearly twice that of the regional average [49,50]. During this period, the population received little to no support from governmental or external actors. This lack of support persisted until 2015, when increasing militarisation garnered more international attention and

surged international humanitarian aid to about four times of what it was in 2014 [13]. As widespread and intense fighting and air strikes continued in the country in parallel to huge increases in humanitarian funding, practical assistance to populations was unevenly distributed as insecurity reduced the reach of aid organisations [51].

Arguably, the level of excess mortality we find in our study could reflect the longstanding deterioration of health situation and non-existence of functional CRVS and health surveillance systems. Hence, when the conflict was declared in 2015, the mortality rate in the year preceding the crisis was not much different from those observed moving forward. Another explanation could also be attributable to limited access of insecure areas, which may have biased the survey results towards a better picture than in reality. While we included all violent deaths by governorate, these metrics may not be reflective of the true count. Such data from conflict countries face a universal bias as regions of high insecurity are often inaccessible to humanitarian actors and surveyors, including for DHS and other UN surveys [53]. More than 50% of the surveys used in this study specifically reported excluding areas due to inaccessibility or insecurity.

Human insecurity in mainland governorates did not correlate positively with higher death rates, despite higher numbers of air strikes and armed attacks in some governorates. Armed violence has severe implications on direct mortality and pervasively disrupts civilian lives. Even when there are no direct hits, both multi-target (e.g., bombs, grenades, shells) and single-target (e.g., targeted killings, snipers) violence profoundly disrupt access to food, markets, healthcare, and other critical commodities by seeding fear of movements. Insecurity has become deeply embedded into the realities of daily life for Yemenis.

In most conflicts, the greatest share of deaths is attributable to indirect causes such as health service breakdowns and food shortages. Direct causes do not usually contribute substantially to overall death rates [44–46]. However, there are isolated occurrences where direct causes do constitute the majority of deaths, as seen in South Sudan from 2013–2018 [54]. In our study, Yemen is another rare example where almost two-thirds of excess deaths were the result of violent causes with the remaining third of deaths due to indirect causes. Almost all governorates experienced sustained air strikes, with some receiving well over a thousand during the conflict (S1 Table). Four sparsely populated governorates, including the Socotra Archipelago, have not endured commensurate levels of air raids [55]. The appreciable increase in humanitarian aid to Yemen following the formal, international recognition of the humanitarian crisis, in conjunction with the already high crude baseline death rates, may explain the observed reduction in indirect deaths.

There are a number of limitations and constraints to consider for this study. We selected a time period for analysis based on available data and the formal recognition of the conflict in Yemen. However, there were appreciable levels of conflict throughout the country prior to the 2015 start date selected for the study. By analysing the post conflict period in the aggregate, we may have missed changes in mortality experience as the intensity of the conflict has changed over time. Both of these decisions are important to consider as alternative explanations for the attenuated differences observed in this study for pre- and conflict crude death rates.

Additionally, across many conflict affected countries, sample units from insecure areas are underrepresented or excluded because of accessibility concern for the safety of personnel [56]. Predictably, this is reflected in surveys undertaken in Yemen, with more than 50% of surveys noting excluding areas due to insecurity or inaccessibility. This lack of access to zones with high violence may bias mortality rates in our, and most other, conflict-event surveys. Further, we acknowledge that the number of surveys from the pre-conflict period limits the robustness of our baseline calculations. Poor data is not unique to Yemen but is rather a persistent problem in conflict-affected countries such as Somalia, Afghanistan, amongst others [57].

Characterising these limitations proves to be both informative to the reality on the ground and provide direction for future work. Rather than discounting information collected under difficult circumstances, these data issues underscore two urgent challenges. First, we need to develop innovative methodologies to collect data in high insecurity conditions, such as using remote sensing techniques. Besson et al (2020) provides a good example of how this technique can be used to calculate excess mortality in Yemen [58]. Second, innovative approaches that build on existing structures should be explored. One such option would be to re-purpose traditional surveillance into sentinel systems in order to bridge the existing gap in data from high insecurity areas. While our study results admittedly cannot be considered definitive because of limited data on which we based our calculation, we are confident that our findings are sufficiently indicative of the reality on the ground.

In summary, massive humanitarian aid to Yemen may have plausibly blunted the humanitarian health crisis in the civil population. The barrage of air strikes and continued ground violence, on the other hand, arguably had the opposite effect on violent causes of death. While the Biden administration pledged to halt US offensive military support by limiting the US's role in supplying weapons to Saudi Arabia and other Gulf States, these promises have not come to fruition. Given the complex history of the conflict and the positions of participating factions, the underlying issues of the conflict are not likely to be resolved soon. Further, there is great potential for increased financial support for conducting air raids over Yemen as the Global North becomes more reliant on oil exports from the Gulf States to compensate for sanctions against the Putin regime following the Russian invasion of Ukraine. As the puppet mastery of geopolitics continues to dictate morbidity and mortality patterns of Yemeni civilians, we must redouble efforts to understand the human health impacts of this sustained crisis. National health and mortality surveys to better understand the full consequences of the conflict on the Yemeni population are needed urgently if and when access to vulnerable populations improves.

## Supporting information

**S1 Fig. Step-by-step explanation of the calculation of excess deaths in Yemen, 2015–2019.** (PDF)

**S1 Table. Distribution of air raids/drone attacks associated direct deaths, and PCDR by governorate, Yemen, January 2015 –December 2019.** The map of controlled area has been sourced from European Council on Foreign Relations website and has been adapted to include number of air raid/drone strikes. It should be considered indicative only and the page can be accessed by the link below: https://ecfr.eu/publication/talking_to_the_houthis_how_europeans_can_promote_peace_in_yemen/. (PDF)

**S2 Table. Survey characteristics, Yemen, 2015–2019.** Characteristics of 91 small-scale mortality surveys in Yemen, 2012–2019. All surveys were reviewed for mortality estimates. This information and more resources can be accessed at: https://www.humanitarianresponse.info/en/operations/yemen/nutrition. (PDF)

**S3 Table. Association between security level and PCDR, Yemen, 2015–2019.** One-way ANOVA test to describe the association between four security levels and PCDR as calculated from 91 small-scale cluster surveys. Security levels were determined by generating security scores and insecurity indices. The insecurity indices were then classified into quantiles for analysis. (PDF)

**S1 Data. Survey data, Yemen, 2015–2019.**
(XLSX)

## Acknowledgments

We would like to thank our Yemeni colleagues from universities in Yemen who have provided invaluable information and advice on interpretation. The circumstances at this time make it difficult for their names to appear in this paper, but their contribution remains key.

The authors would also like to thank Alexandria Williams for her excellent technical editing and Valentin Wathelet for assistance with the extraction of surveys from the CEDAT Repository.

## Author Contributions

**Conceptualization:** Debarati Guha Sapir, Jideofor Thomas Ogbu.

**Data curation:** Jideofor Thomas Ogbu, Jose Manuel Rodriguez-Llanes.

**Formal analysis:** Jideofor Thomas Ogbu.

**Investigation:** Debarati Guha Sapir, Jideofor Thomas Ogbu, Sarah Elizabeth Scales, Maria Moitinho de Almeida, Anne-Francoise Donneau, Anh Diep, Robyn Bernstein, Akram al-Masnai, Jose Manuel Rodriguez-Llanes, Gilbert Burnham.

**Methodology:** Debarati Guha Sapir, Jideofor Thomas Ogbu, Anne-Francoise Donneau, Anh Diep, Jose Manuel Rodriguez-Llanes.

**Project administration:** Debarati Guha Sapir.

**Validation:** Debarati Guha Sapir, Jideofor Thomas Ogbu, Anne-Francoise Donneau, Anh Diep, Jose Manuel Rodriguez-Llanes.

**Writing – original draft:** Debarati Guha Sapir, Jideofor Thomas Ogbu, Sarah Elizabeth Scales, Maria Moitinho de Almeida, Robyn Bernstein.

**Writing – review & editing:** Debarati Guha Sapir, Jideofor Thomas Ogbu, Sarah Elizabeth Scales, Maria Moitinho de Almeida, Anne-Francoise Donneau, Anh Diep, Robyn Bernstein, Akram al-Masnai, Jose Manuel Rodriguez-Llanes, Gilbert Burnham.

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
