## [Decision Letter · Decision Letter 0]

21 Mar 2022

PGPH-D-21-00542

Civil war and death in Yemen: Analysis of SMART survey data, 2012 – 2019

Dear Dr. Scales,

Thank you for submitting your manuscript to PLOS Global Public Health. After careful consideration, we feel that it has merit but does not fully meet PLOS Global Public Health’s publication criteria as it currently stands. Therefore, we invite you to submit a revised version of the manuscript that addresses the points raised during the review process.

Dear authors,

Humble apologies for the late feedback from us, it was complex to find reviewers on the stats/analyses part and I have been in/out emergencie response myself, thanks for your amazing patience and once again apologies for the incredible delay. Please see the comments from the reviewers attached, we hope you can address them. I will make sure we will move forward fast after you have addressed them.

Once again thanks for your submission and good luck with the revisions the manuscript.

Best wishes,

Amrish

We look forward to receiving your revised manuscript.

Kind regards,

Amrish Yashwant Baidjoe, PhD.

Academic Editor

Journal Requirements:

1. Please update your Competing Interests statement. If you have no competing interests to declare, please state: “The authors have declared that no competing interests exist.”

2. Please ensure that you refer to Figure 2 in your text as, if accepted, production will need this reference to link the reader to the figure.

3. Please provide us with a direct link to the base layer of the map used in Figure 4, Figure 5, S1 Figure, and ensure this location is also included in the figure legend. 

Please note that, because all PLOS articles are published under a CC BY license (creativecommons.org/licenses/by/4.0/), we cannot publish proprietary maps such as Google Maps, Mapquest or other copyrighted maps. If your map was obtained from a copyrighted source please amend the figure so that the base map used is from an openly available source.

Please note that only the following CC BY licences are compatible with PLOS licence: CC BY 4.0, CC BY 2.0  and CC BY 3.0, meanwhile such licences as CC BY-ND 3.0 and others are not compatible due to additional restrictions. If you are unsure whether you can use a map or not, please do reach out and we will be able to help you. 

The following websites are good examples of where you can source open access or public domain maps:

Additional Editor Comments (if provided):

Reviewers' comments:

Reviewer's Responses to Questions

**Comments to the Author**

1. Does this manuscript meet PLOS Global Public Health’s publication criteria? Is the manuscript technically sound, and do the data support the conclusions? The manuscript must describe methodologically and ethically rigorous research with conclusions that are appropriately drawn based on the data presented.

Reviewer #1: Yes

Reviewer #2: Yes

2. Has the statistical analysis been performed appropriately and rigorously?

Reviewer #1: I don't know

Reviewer #2: I don't know

3. Have the authors made all data underlying the findings in their manuscript fully available (please refer to the Data Availability Statement at the start of the manuscript PDF file)?

Reviewer #1: No

Reviewer #2: No

4. Is the manuscript presented in an intelligible fashion and written in standard English?

Reviewer #1: Yes

Reviewer #2: No

5. Review Comments to the Author

Reviewer #1: Thank you for the opportunity to review this manuscript, which addresses an important set of questions relating to the impact of conflict on mortality rates in Yemen. This is an under-researched topic, and original health research on Yemen is in any case rare, so the paper is welcome. The results presented are important and the finding regarding the proportionate share of direct and indirect deaths, in particular, is counterintuitive. The caution shown by the authors in interpreting these results in the discussion – particularly by highlighting long running population vulnerabilities in Yemen that long pre-dated the conflict – is to be commended.

I provide some general comments here, and then more detailed comments by section further down.

General comments:

1. I would recommend a statistical review for aspects of the methodology in this paper – I am not qualified to comment on the appropriateness or construction of the Poisson-gamma model used to generate the main results.

2. Of the large team that contributed to this work, only one appears to be Yemeni and none are currently based in country (if the affiliation list is complete). Given that PLOS Global Public Health has set as its mission an effort to address entrenched global health inequalities, and to give voice to those often excluded, this imbalance in contributorship to the paper is somewhat disappointing and something to be addressed for future studies.

3. An important issue in the construction of this analysis is the use of 2015 (the year in which the conflict was internationalised) as a cut-off point to distinguish BCDR from PCDR (as also indicated by table S1). The authors acknowledge this briefly in the discussion in line 373ff. While there was certainly a step-change in the intensity of conflict in that year resulting from international intervention, fighting had been ongoing internally since at least September 2014 (the point at which the Houthis entered Sanaa) and in some localised areas (especially in the North) intermittently but with some intensity to a point well before the beginning of the study time series in 2012. This may contribute to an overestimation of effects especially for governorates such as Hajjah and Al Jawf in the North where direct deaths in the pre-2015 period may have been significant, assuming that the BCDR does not account for violence-related mortality during the 2012-14 period. It may also help explain why the difference in pre-post CDRs is not that great. Do the authors have the data available from the earlier period to comment on this? What effect would incorporation of these numbers have on the TED estimate, in their view?

4. Relatedly, treating the 2015-2019 period as a single block for PCDR calculation may mask large variations in conflict intensity over time. For example, ACLED data show large spikes in reported deaths in 2015 for civilians but markedly lower rates after this point, although this is admittedly not the case for combatants (https://acleddata.com/2019/06/18/yemen-snapshots-2015-2019/). The authors may wish to consider this as a possible explanation for the small different in pre-post CDRs observed in their study, in the discussion.

5. The authors rightly point out the surprisingly high overall contribution of direct mortality to the mortality rate. However, similar results have I think been documented elsewhere (albeit for more focused time periods) and some studies suggest CDRs as high as 1 per 10,000 person-days or more (e.g. https://www.lshtm.ac.uk/south-sudan-full-report). Are the authors able to draw any conclusions based on their data and reading of the wider literature as to how conflict intensity in Yemen compares with other settings?

Specific comments:

6. Line 50-51: please provide the CIs around the BCDR and PCDR estimates here.

7. Line 55: please spell out what PCDR is (you give this in the main text, but the abstract is the first time this abbreviation is seen by the reader).

8. In some areas the phrasing is informal e.g. line 78 - “knee-deep in the largest cholera outbreak…” and could be amended without compromising the central messages that the scale and scope of conflict impact in Yemen has few contemporary precedents.

9. Line 88-99: this section reflects a general tendency to frame the conflict in Yemen as a proxy conflict between two distinct sides, identified in sectarian terms and by relationship to specific external actors. In fact the conflict in Yemen has for some years been a multi-dimensional civil war, primarily driven by internal dynamics. Fighting between groups nominally opposed to the Houthis has at times been just as intense as between the Houthis and others. There is also an important secessionist dimension to the conflict in the South. International alliances for parties in the conflict have shown considerable fluidity over time. There are a number of sources to consider citing here from close observers of Yemen, including work by Bernard Haykel, Helen Lackner and Peter Salisbury.

10. Results – line 240ff: the authors have provided brief details of included surveys in table S2 but it would be helpful to have a clearer view of the content of these. Can the authors comment on the quality of included surveys? In addition, it would be helpful to have an indication of what the reported deaths rates in these individual surveys actually were over time, to (i) build a sense of the spread of the data (including outliers) and (ii) help address point 4 above.

Reviewer #2: Please see comments in the attached file.

This manuscript could be tighter, and it needs another copy edit (which now I'm seeing PLOS won't provide, unfortunately).

I did not review the statistical calculations, outside my area of expertise.

The data really should be accessible to the reader, but were not in a supplemental file as far as I could tell.

6. PLOS authors have the option to publish the peer review history of their article (what does this mean?). If published, this will include your full peer review and any attached files.

**Do you want your identity to be public for this peer review?** For information about this choice, including consent withdrawal, please see our Privacy Policy.

Reviewer #1: **Yes: **Sharif Ismail

Reviewer #2: **Yes: **Amy Hagopian

---

## [Editor Report · Decision Letter 1]

7 Jun 2022

Civil war and death in Yemen: Analysis of SMART survey and ACLED data, 2012 – 2019

Civil war and death in Yemen

PGPH-D-21-00542R1

Dear Ms. Scales,

We are pleased to inform you that your manuscript 'Civil war and death in Yemen: Analysis of SMART survey and ACLED data, 2012 – 2019

Civil war and death in Yemen' has been provisionally accepted for publication in PLOS Global Public Health.

Best regards,

Amrish Yashwant Baidjoe, PhD.

Academic Editor

Dear authors,

Thanks for addressing all comments so dilegently. I am looking forward seeing this manuscript in the the public.

Best wishes,

Amrish